# HIV viral load non-suppression and associated factors among pregnant and postpartum women in rural northeastern South Africa: a cross-sectional survey

Nobubelo Kwanele Ngandu ,[1] Carl J Lombard,[2,3] Thandiwe Elsie Mbira,[1] Adrian Puren,[4] Catriona Waitt,[5,6] Andrew J Prendergast,[7,8] Thorkild Tylleskär,[9] Philippe Van de Perre,[10,11,12] Ameena Ebrahim Goga[1]

For numbered affiliations see end of article.

**Correspondence to**
Dr Nobubelo Kwanele Ngandu;
nobubelo.ngandu@mrc.ac.za

## ABSTRACT

**Objectives** We aimed to measure the prevalence of maternal HIV viral load (VL) non-suppression and assess associated factors, to evaluate progress towards United Nations-AIDS (UNAIDS) targets.

**Design** Cross-sectional study.

**Setting** The eight largest community health centres of Ehlanzeni, a rural district in northeast South Africa.

**Participants** Pregnant women living with HIV (WLHIV) in their third trimester and postpartum WLHIV and their biological infants, recruited equally across all stages of the first 24 months post partum, were included. A sample of 612 mothers participated from a target of 1000.

**Primary outcome measures** The primary outcome was maternal VL (mVL) non-suppression (defined here as mVL >1000 copies/mL). We collected information on antiretroviral use, healthcare visits and sociodemographics through interviews and measured plasma mVL. Descriptive statistics, $\chi^2$ tests and multivariable logistic regression analysis were conducted.

**Results** All mothers (median age: 30 years) were on antiretroviral therapy (ART) and 24.9% were on ART ≤12 months. The prevalence of mVL non-suppression was 14.7% (95% CI: 11.3% to 19.0%), while 13.8% had low-level viraemia (50–1000 copies/mL). Most (68.9%) women had initiated breast feeding and 37.6% were currently breast feeding their infants. Being younger than 25 years (adjusted odds ratio (AOR): 2.6 (95% CI: 1.1 to 6.4)), on first-line ART (AOR: 2.3 (95% CI: 1.1 to 4.6)) and married/cohabiting (AOR: 1.9 (95% CI: 1.0 to 3.7)) were significantly associated with increased odds of mVL non-suppression.

**Conclusions** The prevalence of mVL ≤1000 copies/mL of 85.3% among pregnant and postpartum WLHIV and attending public healthcare centres in this rural district is below the 2020 90–90–90 and 2030 95–95–95 UNAIDS targets. Given that low-level viraemia may also increase the risk of vertical HIV transmission, we recommend strengthened implementation of the new guidelines which include better ART options, improved ART regimen switching and mVL monitoring schedules, and intensified psychosocial support for younger women, while exploring

## Strengths and limitations of this study

► This study provides maternal viral load (mVL) non-suppression data from a rural setting where research of this kind is limited.

► The mVL measurements were conducted using the gold standard whole blood plasma rather than dried blood spots despite the challenging remote settings.

► The postpartum inclusion criterion targeted mothers attending clinics with their biological children and hence could present a sample biased towards biological mothers with better clinic attendance and less representative of mothers who mostly assign child healthcare visits to other caregivers.

► The sample sizes achieved were lower than planned and much lower for the postpartum stages which did not overlap with routine child vaccination schedules, but there was no statistically significant difference in the proportion of mVL non-suppression between the postpartum stages.

district-level complementary interventions, to sustain VLs below 50 copies/mL among all women.

## BACKGROUND

Maternal HIV viral load (VL) during pregnancy and breast feeding is an indicator of the risk of vertical transmission of HIV (ie, mother-to-child transmission of HIV (MTCT)). The HIV test-and-treat approach and universal coverage of daily life-long triple antiretroviral therapy (ART) to achieve viral suppression and prevent MTCT have contributed to substantial decreases in MTCT globally, although the incidence of vertical transmission remains a public health concern.[1] In South Africa, the national average MTCT rate has reduced to <2% at birth, but heterogeneity exists at subregional

levels with much higher intrauterine infections in some districts.[2] In addition, the MTCT risk during the breast-feeding period is critical, because breast milk transmission globally is declining at a slower rate than in utero and intrapartum transmission.[3] Postpartum transmission is perpetuated by incident maternal HIV infection during the breastfeeding period combined with low use of repeat HIV testing among previously uninfected mothers; inadequate monitoring of maternal VL (mVL) and poor postpartum ART adherence.[4–12] Antenatal HIV prevalence has remained around 30% over the past decade in South Africa, and as high as 45% in some districts, making monitoring of mVL a necessity to ensure corrective measures in order to reduce MTCT risk.[13]

In South Africa, although the target of ensuring MTCT below 5% at the end of breast feeding was achieved by 2015, the case rate of <50 new HIV infections per 100 000 live births, to meet global elimination of MTCT (eMTCT) targets, has not been met.[4 14] Until the United Nations-AIDS (UNAIDS) 95–95–95 goals are achieved and sustained across the prevention of mother-to-child transmission of HIV (PMTCT) cascade from pregnancy through to the end of breast feeding, meeting the eMTCT targets could be impossible.[15 16] The first two 95s, that is, 95% of HIV-infected persons knowing their HIV status and 95% of these initiated on ART, are measured during the first antenatal care (ANC) visit and have been periodically reviewed to guide service delivery strengthening in South Africa.[13 17] In South Africa, the UNAIDS 2020 targets of 90% are achieved for these two indicators at ANC entry level but, similar to other low-income and middle-income countries (LMICs) sustaining adequate coverage of HIV diagnoses and ART adherence throughout the antenatal and postnatal period is challenging.[15 18–20] Considering that up to a third of pregnant women living with HIV (WLHIV) only initiate ART after enrolling into ANC, strengthening retention in care to support ART adherence and to monitor VL throughout the PMTCT cascade is critical.[13] Monitoring mVL needs to be a priority as it is the best indicator of ART efficacy, ART adherence and need for enhanced infant postnatal prophylaxis in infants at high risk of HIV acquisition. Despite its importance, the last 95 (95% of all HIV-infected persons on ART should be virally suppressed, ie, have VL below 1000 copies/mL) is not being monitored at a population level along the PMTCT cascade in many LMICs with high HIV burden, including South Africa. This is largely attributed to the financial cost and logistical complications associated with VL measurements. Given the high heterogeneity at subregional levels, routine VL monitoring should be achievable by prioritising hotspot districts currently bearing the highest antenatal and postnatal HIV burden and MTCT risk.

## Study objective

We identified one of the hotspot districts with high maternal HIV prevalence and high MTCT in South Africa and conducted a cross-sectional evaluation of the prevalence of mVL non-suppression (VL >1000 copies/mL) and associated factors during peripartum and postpartum periods, to evaluate progress towards the UNAIDS targets.

## METHODS

### Study design and inclusion criteria

A facility-based cross-sectional study was conducted at the eight largest community healthcare centres (CHC) in Ehlanzeni district, Mpumalanga Province, in rural north-eastern South Africa (bordering Mozambique); seven were in rural localities and one in a peri-urban locality. A district-level sample size was calculated using an assumed mVL non-suppression prevalence of 25% and a precision of 5% at a 90% confidence level.[6 21] The sampling strategy was designed to equally represent different PMTCT cascade milestones across the peripartum and postpartum stages and grouped according to the 2015–2019 PMTCT guidelines for infant antiretroviral prophylaxis and breast feeding.[22] The CHCs were treated as the primary sampling units for data collection. Therefore, the planned study design targeted equal sample sizes of n=200 (25 from each CHC) for each of five groups of WLHIV: third trimester of pregnancy, and four postpartum time points (0–14 weeks, 15–26 weeks, 27–52 weeks (6–11 months) and 53–104 weeks (12–24 months)). The inclusion criteria were woman being in any of these stages, living with HIV and the baby present (for postpartum) on the day of interview. Data collection was conducted over a period of 3 months from mid-September to mid-December 2019 to minimise heterogeneity within the sample.

### Data collection and laboratory measurements

The study was introduced to all female clinic attendees in waiting rooms and interested pregnant and postpartum women were screened for inclusion criteria after receiving their routine care. After written informed consent, the woman was interviewed to collect basic demographic and HIV-related clinical histories from recall and clinic records, and entered directly onto tablets linked to an electronic database (REDCap) securely managed at the host institution.[23] Whole blood was collected from all women for VL measurement and transported to the local laboratory on ice. Plasma was separated within 4–12 hours and stored at −20°C until further analyses at the research laboratory. The HIV-1 VL assays were performed using the Roche COBAS AmpliPrep/COBAS TaqMan HIV-1 quantitative Test, V.2.0 (Roche Diagnostics GmbH, Mannheim, Germany), according to the manufacturer's instructions. The limit of virus detection using this assay was 20 copies/mL at minimum plasma volumes of 1–2 mL.

### Independent variables

#### Sociodemographic factors

Maternal age and body mass index (BMI) were presented as categorical variables. A wide BMI range was observed

and grouped into underweight (13.0–18.4), normal (18.5–24.9), overweight (25.0–29.9), obese (30.0–39.9) and extremely obese (40.0–80.0), with extreme outliers treated as missing. Primary source of income was categorical and average household income was binary defined using the national poverty line household income cutoff of R3200/month. Marital status was either married/cohabiting or other (single, divorced, widow, undisclosed). The highest education achieved was either none, first 7 years of basic education (primary), years 8–12 (secondary), a postsecondary certificate (tertiary-certificate) and postsecondary diploma or higher qualification (Tertiary-diploma or higher). Participant self-reported partner's HIV status was positive, negative or unknown and frequency of condom use was never, sometimes or always.

### ANC variables
Planned pregnancy (yes/no) was self-reported and gestation age at first ANC visit (≤12 weeks, 13–20 weeks, >20 weeks) and number of ANC visits (0–4 or 5–12 visits) were confirmed using clinic records for the most recent pregnancy.

### ART-related factors
These included duration on ART (≤12 months vs >12 months), current ART regimen (first-line regimen vs all others), adherence to ART defined as missed an ART dose during the most recent 7 days (yes/no), ever face challenges with ART adherence (yes/no) and timing of HIV-positive diagnosis (before most recent pregnancy vs during ANC/after). The recommended ART regimens at the time of this study were a fixed-dose combination of TDF + 3TC (or FTC) + EFV with AZT as an option in the case of a contraindication for first-line and TDF (or AZT) + 3TC (or FTC) + LPV/r (or ATV/r) for second-line options.[22] Switch to second line was expected after two consecutive VL >1000 copies/mL 6 months apart on first-line ART or after 2 months of intensive adherence counselling and VL remaining >1000 copies/mL. Switch to third-line regimen was recommended after a VL >1000 was sustained for over 6 months on second line and the drug options were determined through specialist review of drug-resistant mutations. All women were tested for HIV at ANC first visit and every 6 months thereafter.[22]

### Infant-related postnatal factors
Infant age was used to design the study strata. Child health booklets and maternal report were used to obtain infant HIV status at enrolment (positive, negative, unknown), gestational age at birth (≤37 weeks or 38–42 weeks) and birth weight (low birth weight (<2.5 kg) vs higher). Mothers self-reported whether infant was currently breast feeding (yes/no), ever breast fed (yes/no, no response) and currently on anti-HIV prophylaxis (yes/no).

### Dependent variable
The primary outcome measure was mVL non-suppression (VL >1000 copies/mL), in line with the South African

PMTCT guideline criteria for ART regimen management at the time of data collection.[22]

The national PMTCT guidelines practiced during this study recommended 6-monthly VL testing if a woman is virally suppressed (VL ≤1000 copies/mL) or 4–6 weekly if their VL is >1000 copies/mL.[22]

### Statistical analyses
We conducted univariable and multivariable logistic regression analyses to measure the association between the outcome and exposures of interest. Exposure variables were included in the multivariable model only if they had an overall univariable p value <0.2. A significance level of 0.05 in the multivariable analysis was used to indicate a significant association with mVL. Infant-related PMTCT postnatal factors were evaluated for association with mVL separately. $\chi^2$ tests were used to assess associations between categorical variables and descriptive statistics were done.

In all tests conducted, a stratified survey-based analysis was used to combine the five study groups into a single analysis. As per study design, the study groups were specified as the strata and the CHCs were treated as primary sampling units, in the survey structure for data analysis. In addition, all proportions and ORs were adjusted (weighted) for sample size attained by specifying survey weights in the survey structure and in all analysis. The survey weights were calculated by taking the inverse of sample-size realisation proportion, where proportions were calculated using the primary sampling unit (facility-level) sample target of n=25 within each study group stratum. Study group stratification was not applied in the supplementary descriptive analyses of each study group separately.

The Strengthening the Reporting of Observational Studies in Epidemiology checklist for observational cross-sectional studies was followed (online supplemental file 1).

### Patient and public involvement
No patient involved.

## RESULTS
During the 3-month data collection period, interview data were successfully collected from 667 WLHIV, translating to sample size realisation of 66% overall—within group ascertainment was 88% for third trimester group (n=176), 64% for 0–14 weeks postpartum group (n=128), 38% for 15–26 weeks postpartum group (n=75), 62% for 27–52 weeks postpartum group (n=123) and 55% for 53–104 weeks postpartum group (n=110). The precision for these achieved samples remained close to 0.05 and was 0.05, 0.0514, 0.068, 0.0526 and 0.056, respectively. Very few mother–infant pairs in the 15–26 weeks postpartum group, a period with no routine vaccination schedules, were present in the clinics. VL results were available for 612 (91.8%)/667 interviewed women; 55 (8.2%) samples

had insufficient plasma. The 55 women with insufficient plasma were excluded from the analysis.

### Description of the study population
#### Socio-demogaphics and ANC

The median age was 30 years (IQR: 26–35), 8.6% and 58.1% had 0–7 years and 8–12 years of completed education, respectively, while the remaining third had some tertiary education. Most (59.6%) came from households with average earnings below the poverty income line (<R3200 per month) (table 1). A quarter of the participants were employed, a third relied on government grants while the remainder depended on another individual (parent/relative/spouse/partner) as their main source of income; 43.5% women were married or cohabiting; 46.4% knew they were in a concordant HIV-positive sexual relationship, while 35.4% did not know their partner's HIV status. Half of the participants had an unplanned pregnancy and around two thirds had their first ANC visit during the first 12 weeks of gestation and achieved at least five ANC visits.

#### ART-related factors

Over 70% of participants were diagnosed with HIV before the current/recent pregnancy. All 612 women were already on ART at study enrolment and three quarters had been on ART for more than 12 months. Most (80.2%) women were still on their first-line regimen. Although self-reported levels of adherence to ART during the most recent 7 days were very high (95.2%), 38.0% reported ever experiencing challenges which potentially interfere with adherence to ART. The commonly cited barriers were time inconvenience (22.8%), non-disclosure of HIV status (11.7%), ART side effects (5.4%) and lack of support from family/partner (3.2%).

#### Infant-related postnatal factors

Close to 70% of the infants were ever breast fed and 37.6% were currently breast feeding. Overall, 29% infants were still taking prophylaxis, higher than the 0–14 weeks postnatal sample proportion of 23% expected to be on prophylaxis. At enrolment, 79.2% of mothers knew their infants to be HIV-negative and 2.1% to be HIV-positive, while 18.7% did not know the infant's HIV status. Over 80% of infants were born full-term and with weight ≥2.5 kg.

#### Prevalence of mVL non-suppression

mVL ranged between undetectable and 557 500 copies/mL. The median of detectable values was 138 copies/mL (IQR: 35–3660 copies/mL). Overall, 14.7% (95% CI: 11.3 to 19.0) of the women had mVL non-suppression while 13.8% (95% CI: 10.2 to 18.4) had low-level viraemia (VL 50–1000 copies/mL) and 71.5% (95% CI: 63.7 to 78.2) had policy-defined undetectable (<50 copies/mL) mVL.[22] VL non-suppression by study group ranged between 6.9% and 17.5%, with all groups except the 15–26 weeks postpartum group, showing a prevalence of 15.0% or higher (table 2). Although mVL non-suppression was lower for

**Table 1** Summary of study population characteristics

| | n | % |
|---|---|---|
| All | 612 | 100 |
| **Study group** | | |
| Third trimester | 176 | 28.8 |
| 0–14 weeks postpartum | 128 | 20.9 |
| 15–26 weeks postpartum | 75 | 12.2 |
| 27–52 weeks postpartum | 123 | 20.1 |
| 53–104 weeks postpartum | 110 | 18.0 |
| **Sociodemographics and ANC** | | |
| Age in years | | |
| 15–24 | 105 | 17.8 |
| 25–34 | 351 | 57.2 |
| 35–46 | 156 | 25.0 |
| BMI* | | |
| 13.0–18.4 | 20 | 3.6 |
| 18.5–24.4 | 200 | 35.4 |
| 25.0–29.9 | 175 | 28.6 |
| 30.0–39.9 | 173 | 27.1 |
| 40.0–80.0 | 39 | 5.3 |
| Education | | |
| None | 8 | 1.2 |
| Primary (1–7 years) | 44 | 7.4 |
| Secondary (8–12 years) | 347 | 58.1 |
| Tertiary-certificate | 155 | 23.3 |
| Tertiary-diploma or higher | 58 | 10.0 |
| Married/cohabiting | | |
| No | 362 | 56.5 |
| Yes | 250 | 43.5 |
| Income source* | | |
| Employed | 163 | 24.2 |
| Spouse/partner | 187 | 32.5 |
| Parent/relative | 90 | 14.2 |
| Grant | 166 | 29.1 |
| Household gross income/month | | |
| >R3200 | 262 | 40.4 |
| R3200 or less/none | 349 | 59.6 |
| Partner's HIV status* | | |
| Negative | 110 | 18.1 |
| Positive | 279 | 46.4 |
| Do not know | 222 | 35.4 |
| Condom use frequency* | | |
| Never | 52 | 7.2 |
| Sometimes | 226 | 35.8 |
| Always | 330 | 57.0 |
| Planned pregnancy | | |
| No | 330 | 52.3 |

Continued

**Table 1** Continued

| | n | % |
|---|---|---|
| Yes | 282 | 47.7 |
| **Gestation at ANC-1 visit** | | |
| ≤12 weeks | 379 | 65.8 |
| 13–20 weeks | 165 | 23.9 |
| >20 weeks | 68 | 10.3 |
| **Number of ANC visits*** | | |
| 0–4 visits | 210 | 32.2 |
| 5–12 visits | 401 | 67.8 |
| **PMTCT exposure variables** | | |
| **Timing of HIV-positive result** | | |
| Before pregnancy | 448 | 73.1 |
| At ANC-1 or after | 164 | 26.9 |
| **Time since ART initiation** | | |
| >12 months | 442 | 85.1 |
| ≤12 months | 170 | 24.9 |
| **Current ART regimen*** | | |
| First line | 491 | 80.2 |
| Second/third line or unknown | 119 | 19.8 |
| **Missed an ART dose last 7 days** | | |
| No | 582 | 95.2 |
| Yes | 30 | 4.8 |
| **Facing any ART adherence challenges** | | |
| No | 376 | 62.0 |
| Yes | 236 | 38.0 |
| **Infant-related factors (postpartum sample only)** | **436** | **100** |
| **Infant ever breast fed*** | | |
| No | 146 | 31.1 |
| Yes | 289 | 68.9 |
| **Infant currently breast feeding** | | |
| Yes | 163 | 37.6 |
| No | 126 | 31.0 |
| Chose not to disclose | 147 | 31.4 |
| **Infant currently on ARV prophylaxis*** | | |
| No | 293 | 71.0 |
| Yes | 140 | 29.0 |
| **Infant HIV status at enrolment** | | |
| Negative | 345 | 79.2 |
| Positive | 10 | 2.1 |
| Unknown | 81 | 18.7 |
| **Gestational age at birth*** | | |
| ≤37 weeks | 67 | 18.7 |
| 38–42 weeks | 367 | 81.3 |

Continued

**Table 1** Continued

| | n | % |
|---|---|---|
| **Infant birth weight*** | | |
| Birth weight ≥2.5 kg | 392 | 90.6 |
| Low birth weight (<2.5 kg) | 43 | 9.4 |

*Denominator less than N due to missing responses.
ANC, antenatal care; ART, antiretroviral therapy; ARV, antiretroviral; BMI, body mass index; PMTCT, prevention of mother-to-child transmission of HIV.

the participants in the 15–26 weeks group, there was no statistically significant difference across PMTCT stages ($\chi^2$ p=0.284).

### Summary of study population characteristics by mVL non-suppression

The prevalence of mVL non-suppression differed significantly by source of income (p=0.018), ART duration (p=0.029) and ART regimen (p=0.016) (table 2). The proportion of VL >1000 copies/mL was higher among women who depended on a parent/relative or spouse/partner as their main source of income, were on ART for ≤12 months, or were on first-line ART.

Weak evidence of differences in the prevalence of mVL non-suppression were also observed when women were grouped by maternal age (p=0.054), BMI (p=0.075), household income level (p=0.097), knowledge of partner's HIV status (p=0.081) and timing of first HIV-positive result (p=0.086). The proportion of viral non-suppression tended to be higher among women who were younger than 25 years of age, had BMI between 18.5 and 29.9, came from households with average income >R3200, aware that their partner was HIV-positive or did not know their partner's HIV status or received the first HIV-positive diagnoses after enrolling into ANC for the most recent pregnancy.

Infant-related postnatal PMTCT characteristics did not appear to distinguish women by prevalence of VL non-suppression (table 2).

The distribution of mVL non-suppression by participant characteristics within each study group separately is presented in the supplementary data (online supplemental file 2).

### Factors associated with mVL non-suppression

In univariable analyses to identify factors associated with mVL non-suppression among all women, independent variables with an overall OR p value <0.2 and included in the multivariable logistic regression model were ART duration, ART regimen maternal age, BMI, marital status, household income level, partner's HIV status and condom use (table 3). The 'timing of first HIV positive result' variable was excluded from the multivariable model because it showed collinearity with ART duration (variance inflation factor of the interaction term was above 5, ie, 6.3). The OR p values of other variables including that for the

**Table 2** Summary of study population characteristics by maternal VL non-suppression

| | | VL ≤1000 copies/mL | | VL >1000 copies/mL | | P value |
|---|---|---|---|---|---|---|
| | | n | % (95% CI) | n | % (95% CI) | |
| **All** | | 526 | 85.3 (81.6 to 88.3) | 86 | 14.7 (11.7 to 18.4) | |
| Study group | Third trimester | 147 | 82.9 (76.9 to 87.6) | 29 | 17.1 (12.4 to 23.1) | 0.284 |
| | 0–14 weeks postpartum | 112 | 85.0 (76.5 to 90.8) | 16 | 15.0 (9.2 to 23.5) | |
| | 15–26 weeks postpartum | 70 | 93.1 (82.5 to 97.4) | 5 | 6.9 (2.6 to 17.5) | |
| | 27–52 weeks postpartum | 102 | 82.5 (71.1 to 90.1) | 21 | 17.5 (10.0 to 28.9) | |
| | 53–104 weeks postpartum | 95 | 83.6 (74.5 to 89.9) | 15 | 16.4(10.1 to 25.2) | |
| **Maternal sociodemographics and ANC** | | | | | | |
| Age in years | 15–24 | 82 | 76.5 (65.5 to 84.8) | 23 | 23.5 (15.2 to 34.5) | 0.054 |
| | 25–34 | 305 | 86.2 (80.3 to 90.5) | 46 | 13.8 (9.5 to 19.7) | |
| | 35–46 | 139 | 89.5 (82.8 to 93.8) | 17 | 10.5 (6.2 to 17.2) | |
| BMI | 13.0–18.4 | 17 | 88.1 (63.0 to 97.0) | 3 | 11.9 (3.0 to 37.0) | 0.075 |
| | 18.5–24.9 | 171 | 84.1 (76.5 to 89.6) | 29 | 15.9 (10.4 to 23.5) | |
| | 25.0–29.9 | 140 | 79.4 (70.1 to 86.4) | 35 | 20.6 (13.6 to 29.9) | |
| | 30.0–39.9 | 157 | 90.7 (86.6 to 93.7) | 16 | 9.3 (6.3 to 13.4) | |
| | 40.0–80.0 | 36 | 92.5 (82.6 to 97.0) | 3 | 7.5 (3.0 to 17.4) | |
| Education | None | 6 | 66.5 (31.0 to 90.0) | 2 | 33.5 (10.3 to 69.0) | 0.333 |
| | Primary (1–7 years) | 41 | 94.5 (81.2 to 98.5) | 3 | 5.5 (1.5 to 18.8) | |
| | Secondary (8–12 years) | 297 | 84.0 (78.5 to 88.4) | 50 | 16.0 (11.6 to 21.5) | |
| | Tertiary-certificate | 131 | 85.8 (78.6 to 90.9) | 24 | 14.2 (9.1 to 21.4) | |
| | Tertiary-diploma/ higher | 51 | 86.7 (70.5 to 94.7) | 7 | 13.3 (5.3 to 29.5) | |
| Married/cohabiting | No | 319 | 87.7 (83.8 to 90.9) | 43 | 12.3 (9.1 to 16.3) | 0.122 |
| | Yes | 207 | 82.1 (74.7 to 87.7) | 43 | 17.9 (12.3 to 25.3) | |
| Income source* | Employed | 145 | 88.9 (80.6 to 94.0) | 18 | 11.1 (6.0 to 19.4) | **0.018** |
| | Spouse/partner | 153 | 80.9 (72.6 to 87.2) | 34 | 19.1 (12.8 to 27.4) | |
| | Parent/relative | 72 | 75.9 (62.8 to 85.4) | 18 | 24.1 (14.6 to 37.2) | |
| | Grant | 150 | 91.3 (85.5 to 94.9) | 16 | 8.7 (5.1 to 14.5) | |
| Household monthly gross income | >R3200 | 215 | 82.0 (75.9 to 86.8) | 47 | 18.0 (13.2 to 24.1) | 0.097 |
| | R3200 or less/none | 310 | 87.5 (82.8 to 91.1) | 39 | 12.5 (9.0 to 17.2) | |
| Partner's HIV status | Negative | 98 | 91.6 (85.8 to 95.2) | 12 | 8.4 (4.8 to 14.2) | 0.081 |
| | Positive | 243 | 85.0 (79.1 to 89.4) | 36 | 15.0 (10.6 to 20.9) | |
| | Do not know | 184 | 82.4 (76.4 to 87.2) | 38 | 17.6 (12.8 to 23.6) | |
| Condom use frequency* | Never | 43 | 85.9 (75.4 to 92.3) | 9 | 14.1 (7.7 to 24.6) | 0.235 |
| | Sometimes | 200 | 88.6 (82.6 to 92.8) | 26 | 11.4 (7.2 to 17.4) | |
| | Always | 280 | 83.2 (77.5 to 87.6) | 50 | 16.9 (12.4 to 22.5) | |
| Planned pregnancy | No | 289 | 86.2 (81.0 to 90.2) | 41 | 13.8 (9.8 to 19.0) | 0.611 |
| | Yes | 237 | 84.3 (77.9 to 89.1) | 45 | 15.7 (10.9 to 22.1) | |
| Gestational age at ANC-1 visit | ≤12 weeks | 335 | 86.6 (81.8 to 90.3) | 44 | 13.4 (9.7 to 18.2) | 0.443 |
| | 13–20 weeks | 136 | 83.2 (76.2 to 88.5) | 29 | 16.8 (11.6 to 23.8) | |
| | >20 weeks | 55 | 81.9 (70.9 to 89.4) | 13 | 18.1 (10.6 to 29.1) | |

Continued

**Table 2** Continued

| | | VL ≤1000 copies/mL | | VL >1000 copies/mL | | |
|---|---|---|---|---|---|---|
| | | n | % (95% CI) | n | % (95% CI) | P value |
| Number of ANC visits* | 0–4 visits | 180 | 85.1 (79.5 to 89.3) | 30 | 14.9 (10.7 to 20.5) | 0.921 |
| | 5–12 visits | 345 | 85.4 (81.1 to 88.8) | 56 | 14.6 (11.2 to 18.9) | |
| **ART-related factors** | | | | | | |
| Timing of HIV-positive result | Before pregnancy | 394 | 87.3 (82.6 to 90.9) | 54 | 12.7 (9.1 to 17.4) | 0.086 |
| | At ANC-1 or after | 132 | 79.8 (71.1 to 86.4) | 32 | 20.2 (13.6 to 28.9) | |
| Time since ART initiation | >12 months | 392 | 87.5 (83.0 to 90.9) | 50 | 12.5 (9.1 to 17.0) | **0.029** |
| | ≤12 months | 134 | 78.6 (70.3 to 85.0) | 36 | 21.4 (15.0 to 29.7) | |
| Current ART regimen* | second/3rd line or unknown | 107 | 92.3 (85.9 to 95.9) | 12 | 7.7 (4.1 to 14.2) | **0.016** |
| | First-line | 417 | 83.5 (79.5 to 86.9) | 74 | 16.5 (13.1 to 20.5) | |
| Missed an ART dose last 7 days | No | 501 | 85.7 (82.1 to 88.6) | 81 | 14.4 (11.4 to 17.9) | 0.294 |
| | Yes | 25 | 78.1 (56.0 to 90.9) | 5 | 21.9 (9.1 to 44.1) | |
| Facing any ART adherence challenges | No | 331 | 86.3 (81.3 to 90.0) | 45 | 13.8 (10.0 to 18.7) | 0.551 |
| | Yes | 195 | 83.7 (75.8 to 89.4) | 41 | 16.3 (10.6 to 24.2) | |
| **Infant related factors (postpartum sample only)** | | | | | | |
| | All | 379 | 85.8 (81.4 to 89.4) | 57 | 14.2 (10.6 to 18.6) | |
| Infant HIV status at enrolment | Negative | 296 | 85.4 (80.3 to 89.4) | 49 | 14.6 (10.6 to 19.7) | 0.882 |
| | Positive | 9 | 88.3 (50.1 to 98.3) | 1 | 11.7 (1.7 to 49.9) | |
| | Unknown | 74 | 87.5 (72.6 to 94.9) | 7 | 12.5 (5.1 to 27.5) | |
| Infant currently on ARV prophylaxis* | No | 258 | 87.0 (81.8 to 90.0) | 35 | 13.0 (0.1 to 18.2) | 0.326 |
| | Yes | 119 | 83.5 (75.3 to 89.4) | 21 | 16.5 (10.7 to 24.7) | |
| Infant ever breast fed* | No | 124 | 82.3 (72.7 to 89.1) | 22 | 17.8 (11.0 to 27.3) | 0.231 |
| | Yes | 254 | 87.4 (82.5 to 91.0) | 35 | 12.6 (9.0 to 17.5) | |
| Infant currently breast feeding | Yes | 143 | 87.5 (81.4 to 91.8) | 20 | 12.5 (8.2 to 18.6) | 0.427 |
| | No | 111 | 87.2 (79.9 to 92.1) | 15 | 12.8 (7.9 to 20.1) | |
| | Chose not to disclose | 125 | 82.5 (73.0 to 89.2) | 22 | 17.5 (10.8 to 27.0) | |
| Gestational age at birth* | ≤37 weeks | 56 | 83.1 (70.9 to 90.8) | 11 | 16.9 (9.2 to 29.1) | 0.466 |
| | 38–42 weeks | 322 | 86.9 (81.9 to 90.7) | 45 | 13.1 (9.3 to 18.1) | |
| Infant birth weight* | Birth weight ≥2.5 kg | 345 | 86.6 (82.3 to 89.9) | 47 | 13.4 (10.1 to 17.7) | 0.151 |
| | Low birth weight | 33 | 78.4 (61.9 to 89.1) | 10 | 21.6 (10.9 to 38.1) | |

P values are from a $\chi^2$ test.

Significant p values <0.05 are in boldface font.

*Denominator less than n due to missing responses.

ANC-1, antenatal care first visit; ART, antiretroviral therapy; ARV, antiretroviral; BMI, body mass index; VL, viral load.

study group stratification were >0.2 and hence excluded from the adjusted model. All infant-related postnatal variables had OR p values >0.2 and an adjusted model analyses was not conducted (table 4).

In the adjusted model (table 3), the odds of viral non-suppression were significantly increased among women who were younger than 25 years compared with those aged 35–46 years (adjusted odds ratio (AOR): 2.6 (95% CI: 1.1 to 6.4), p=0.037), were on first-line ART regimen (AOR: 2.3 (95% CI: 1.1 to 4.6), p=0.026) and were married or cohabiting (AOR: 1.9 (95% CI: 1.0 to 3.7),

p=0.042). Weak associations were observed for increased odds of viral non-suppression among women who initiated ART within the most recent 12 months (AOR: 1.7 (95% CI: 0.8 to 3.6), p=0.126) or did not know their male partner's HIV status (AOR: 2.1 (95% CI: 0.9 to 4.8), p=0.080).

Extremely obese BMI of ≥40 was significantly associated with reduced odds of viral non-suppression compared with normal BMI of 18.5–24.9 (AOR: 0.3 (95% CI: 0.1 to 0.9), p=0.028). There was no difference between normal BMI and either underweight, overweight or obese BMI.

**Table 3** Factors associated with maternal viral load non-suppression among all women

| | OR (95% CI) | P value | AOR (95% CI) | P value |
|---|---|---|---|---|
| Time since ART initiation | | | | |
| >12 months | Ref | | | |
| ≤12 months | 1.9 (1.1 to 3.4) | **0.031** | 1.7 (0.8 to 3.6) | 0.126 |
| Current ART regimen | | | | |
| Second/third line or unknown | Ref | | | |
| First line | 2.4 (1.2 to 4.8) | **0.018** | 2.3 (1.1 to 4.6) | **0.026** |
| Age in years | | | | |
| 35–46 | Ref | | | |
| 25–34 | 1.4 (0.7 to 2.8) | 0.376 | 1.1 (0.5 to 2.4) | **0.733** |
| 15–24 | 2.6 (1.1 to 6.2) | **0.029** | 2.6 (1.1 to 6.4) | **0.037** |
| BMI | | | | |
| 18.5–24.9 | Ref | | | |
| 13.0–18.4 | 0.7 (0.1 to 3.8) | 0.689 | 0.8 (0.2 to 4.4) | 0.8146 |
| 25.0–29.9 | 1.4 (0.7 to 2.8) | 0.379 | 1.4 (0.6 to 2.9) | 0.415 |
| 30.0–39.9 | 0.5 (0.3 to 1.0) | 0.051 | 0.5 (0.3 to 1.1) | 0.089 |
| 40.0–80.0 | 0.4 (0.2 to 1.2) | 0.112 | 0.3 (0.1 to 0.9) | **0.028** |
| Married/cohabiting | | | | |
| No | Ref | | | |
| Yes | 1.6 (0.8 to 2.8) | 0.124 | 1.9 (1.0 to 3.7) | **0.042** |
| Partner's HIV status | | | | |
| Negative | Ref | | | |
| Positive | 1.9 (0.9 to 4.3) | 0.101 | 1.9 (0.9 to 4.1) | 0.096 |
| Do not know | 2.3 (1.2 to 4.7) | **0.020** | 2.1 (0.9 to 4.8) | 0.080 |
| Condom use frequency | | | | |
| Never | Ref | | | |
| Sometimes | 0.8 (0.3 to 1.9) | 0.579 | 0.8 (0.3 to 2.0) | 0.628 |
| Always | 1.2 (0.6 to 2.6) | 0.583 | 1.1 (0.6 to 2.1) | 0.769 |
| Household gross income/month | | | | |
| >R3200 | Ref | | | |
| R3200 or less/none | 0.6 (0.3 to 1.1) | 0.099 | 0.6 (0.4 to 1.0) | 0.073 |

Significant p values <0.05 are in boldface font.
ANC-1, antenatal care first visit; AOR, adjusted odds ratio; ART, antiretroviral therapy; BMI, body mass index; OR, Odds Ratio.

## DISCUSSION

The prevalence of mVL <1000 copies/mL in this rural South African district was estimated at 85.3%, and is still below the UNAIDS 2020 and[16] targets of 90% and 95%, respectively, among persons on ART.[15 16] The observed VL suppression prevalence is higher than the 2017 South Africa national antenatal survey estimate of 79.5%, indicating that the district is comparatively performing well.[24] The postpartum study inclusion criteria of biological mothers could be biasing results towards women who frequently attend healthcare services with their biological children, and possibly underestimating mVL non-suppression prevalence without those women who assign child postnatal care activities to other caregivers. The observed prevalence is comparable to PMTCT population in other African countries such as Malawi (84%–88%), but is lower than that reported in an urban setting (91%) within South Africa, further supporting the existing interdistrict disparities in PMTCT performance in-country.[6 19 25] These subnational variations highlight the need for district-level surveys as opposed to provincial-level surveys which mask existing heterogeneity at district-level.

Even though the cross-sectional study design limited our understanding of causality, the significantly increased odds of having unsuppressed VL among women on first-line ART regimens could be related to the observed high proportion of unsuppressed VL among women who were on ART for ≤12 months. Alternatively, it could be due to the dominance of efavirenz-based first-line regimen which has been shown to be associated with delayed viral suppression in this population[26 27] or a high population-level risk of drug resistance

**Table 4** Association between infant-related postnatal factors and maternal viral load non-suppression

|  | OR (95% CI) | P value |
|---|---|---|
| **All** | | |
| **Gestational age at birth** | | |
| ≤37 weeks | Ref | |
| 38–42 weeks | 0.7 (0.3 to 1.7) | 0.467 |
| **Infant birth weight** | | |
| Birth weight ≥2.5 kg | Ref | |
| Low birth weight | 1.8 (0.8 to 4.0) | 0.156 |
| **Infant HIV status at enrolment** | | |
| Negative | Ref | |
| Positive | 0.8 (0.1 to 5.8) | 0.797 |
| Unknown | 0.8 (0.3 to 2.4) | 0.736 |
| **Infant currently on ARV prophylaxis** | | |
| No | Ref | |
| Yes | 1.3 (0.4 to 2.4) | 0.327 |
| **Infant ever breast fed** | | |
| No | Ref | |
| Yes | 0.7 (0.3 to 1.3) | 0.233 |
| **Infant currently breast feeding** | | |
| Yes | Ref | |
| No | 1.0 (0.5 to 2.0) | 0.935 |
| Chose not to disclose | 1.4 (0.7 to 3.0) | 0.265 |

ARV, antiretroviral; OR, Odds Ratio.

previously observed in South African rural populations,[28] or a combination of these factors. The new guidelines currently being rolled-out offer pregnant women a dolutegravir-based regimen as a first-line choice and alternative to efavirenz regimen as part of the new efforts to improve VL suppression.[29] The extremely high self-reported ART well-adherent (95.2%) group likely includes a desirability bias, but other studies in African settings have shown reliability of self-reported ART adherence including its association with mVL.[25 30] Therefore, we conclude that the disproportionately high prevalence of unsuppressed mVL in the group on first-line ART regimen implies a combination of challenges with adherence among those recently initiated on ART and first-line regimen failure. Findings from Zimbabwe showed that being on second-line ART regimen vs first-line was associated with higher odds of viral suppression.[31] It is possible that regimen intervention or shift to second-line ART was delayed among some women. This is further supported by that three quarters of the study sample had been on ART for more than a year, and 90% had their first ANC visit early before 20 weeks gestation at which point over 70% already knew their HIV-positive status and were on ART.

One likely underlying driver of high mVL association with first-line ART is the 6-months long window recommended for following up those with low-level viraemia (50–1000 copies/mL), who however, are at risk of higher VL during this period. The 6 months window of routine follow-up visit when VL ≤1000 copies/mL, was too long considering that it included low-level viraemia which may account for over 40% of early MTCT in some settings.[3 6 22 25] Given the new policies emphasising targeting complete viral suppression of <50 copies/mL as opposed to VL <1000 copies/mL, our results of 71.5% complete viral suppression shows that more work remains to be done.[29] The newly revised 2019 South African PMTCT guidelines have shortened the 6-months window to 8–10 weeks when VL is 50–999 copies/mL to address the problem of delayed VL and ART management, and have provided tips for the healthcare worker to understand barriers to ART adherence such as non-disclosure along with patient care interventions.[29] In addition, those newly initiated on ART are reviewed at 3 months regardless of baseline VL level, after which they are followed according to the universal schedule. These introduced shorter follow-up windows are expected to improve VL management going forward, particularly for those with recent ART initiation.

Periodic monitoring, evaluation and strengthening of these new guidelines as well as any outstanding service-delivery or user-related barriers is imperative. Client-related barriers and long turnaround times for availability of VL results are examples of long-standing challenges that interfere with retention in care and timely follow-up management of VL and ART in LMIC settings.[7 32–35] In a 2018 evaluation of this district and five others, for example, we found that on average at district-level, between 37% and 100% of clinics delay (>7 days) returning VL results to PMTCT clients after the clinic has received them from the laboratory.[36] Given that the most cited potential barrier to ART adherence in this current study was time inconvenience, we hypothesise that this factor contributed to women delaying clinic attendance for their follow-up visits. At present the country is implementing an electronic monitoring system called E-labs to shorten the turnaround times along the VL processing pipeline from blood draw at clinics until availability of results at the testing clinic.[37] Mobile phone messaging directly to patients, such as MomConnect, should be incorporated into E-labs to shorten time to clinic–patient contact.[38] Point of care (POC) VL testing which is not common in this district should also be considered, given strong evidence for improved turnaround times when using POC testing.[39–41] Alternatively, long-acting injectable antiretroviral therapeutics could ultimately reduce the burden of frequent clinic visits and daily pill adherence challenges, and thus studies developing these need to be strengthened earnestly.[11]

The observed association of mVL non-suppression with young women aged 15–24 years is not unexpected. Earlier (2010–2014) studies, including a national PMTCT survey secondary analysis showed that adolescent PMTCT clients in South Africa are at higher risk of undesirable clinical and healthcare uptake outcomes.[42 43] Findings remain similar in the more recent (2016–2017) national representative data in South Africa, where younger age

was significantly associated with failure to achieve undetectable VL.[24][44] The revised PMTCT guidelines have included a guidance for adolescent-aware recommendations during antenatal and postpartum care. The guidance advises healthcare workers to identify psychosocial stressors and provide referrals for professional interventions, and emphasises intensive adherence support such as enrolment into peer-led support groups.[29] Based on a systematic review of impactful interventions for improving healthcare uptake and adherence, family-centred initiatives could also be investigated for improving HIV health among pregnant and breastfeeding adolescents in this setting.[45]

The high prevalence of unsuppressed VL observed among women who depended on another person including a spouse/partner for income aligns with the significantly increased odds among those who were married or cohabiting. The particularly high proportion (35%) of women who do not know their male partner's HIV status is also concerning and could be another underlying contributing factor. HIV disclosure to sexual partners, partner relations and partner involvement as determinants of utilisation of HIV services and HIV health outcomes among pregnant and postnatal women have been long-time challenges in African settings.[46–49] Non-disclosure of HIV status was the second highest cited potential barrier to ART adherence (11.7%) in this study. Data from pregnant women from the same province as this study district, collected 7 years ago, indicated non-disclosure of HIV status as a key factor in poor ART adherence.[50] Disclosure to a partner has been associated with better HIV care outcomes in mothers and their babies in Rwanda.[51] An earlier study in Zimbabwe observed reduced uptake of HIV care among eligible PMTCT clients living with a male partner.[49] Given challenges involving male partners and particularly for younger women, the adolescent-aware recommendations in the current PMTCT guidelines do emphasise on sexual reproductive health counselling during ANC and identification of sociocontextual barriers such as abusive relationships to offer counselling referrals and ensure a safe and healthy home environment.[29] More work is however needed in this area.

The significant association between extreme BMI and reduced odds of VL non-suppression is not well understood and needs further investigation. A previous HIV vaccine efficacy trial reported an association between overweight/obese BMI and suppressed vaccine-induced immune response against HIV in a South African population.[52] Other separate studies on HIV-positive persons from different countries, initiated on antiretroviral treatment, found high BMI to be associated with better CD4 count recovery.[53–55] Whether our findings are simply due to the increased prevalence of obesity in South Africa (with women ranking in the highest burdened countries), and/or increased weight gain due to antiretrovirals as previously implied or there is an underlying biological mechanism, is unclear and needs further investigation.[56–58]

### Limitations

Although the 3-month recruitment period was expected to achieve the desired sample size, assuming 50% participation rate, the realised sample sizes were not as high as planned. However, there was no statistically significant heterogeneity in the primary outcome between the study groups, making the survey-weighted results reliable. The precision achieved by the realised sample size remained acceptable. The reduced sample size in the postpartum groups could have resulted from the study's inclusion criterion of biological mother–infant pairs. It is common practice locally to assign child healthcare visits to other caregivers. The 15–26 weeks postpartum study group had the lowest sample size which we attribute to lack of routine childhood vaccination points during this child growth period. The study by recruiting in facilities also excluded clients who had defaulted from care. Although the findings of our sample could have introduced a selection bias, the VL suppression estimates lie within the range of other estimates reported in the country. Desirability bias from self-reported adherence to ART and condom use and knowing of male partner's HIV status, cannot be ruled out completely. Use of a 7-day recall for adherence to ART, although limiting recall bias, might suffer from low representativeness of the entire period since the last clinic contact. The possible influence of specific ART drugs could not be assessed due to lack of data on specific drug combinations used. This, however, may have minor effects given that options for the first-line regimen were limited to efavirenz-based regimens, hence not much diversity in specific drugs used.[22]

The cross-sectional study design introduces inherent limitations. Causality could not be inferred in this cross-sectional study design and a prospective cohort approach would be needed to confirm some of the associations. An example of the limitation of a cross-sectional design in this context is not being able to understand the timing of infant HIV testing and the relationship between infant HIV-positivity versus mVL, given there is no information on duration of non-suppression in the mother against attrition in care, interruptions in ART intake and feeding practices.

The sample was designed to measure prevalence of VL >1000 copies/mL and the data were not sufficient to investigate differences between VL stages within the 50–1000 copies/mL range to fully inform the newly implemented guidelines. The result of VL=50–1000 copies/mL is only reported descriptively. Virological failure and treatment failure could not be confirmed with certainty given only one VL measurement done.

### CONCLUSIONS

The UNAIDS 2020/2030 targets for viral suppression among PMTCT clients in this district have not been achieved. This study highlights that sociodemographic factors (maternal age, BMI and marital/cohabiting status) and type of ART regimen disproportionately influence

mVL suppression. While the role of BMI requires further investigation, recommendations outlined in the current South African PMTCT guidelines have potential to mitigate other identified factors through, among other revisions, the improved schedules for monitoring VL and ART regimen switching, better alternatives for ART drugs and targeted psychosocio support for younger women with consideration of marital/male partner influence. Given that low-level viraemia may also increase the risk of vertical HIV transmission, it is imperative to periodically monitor, evaluate and strengthen the implementation of these guidelines alongside complementary district-level interventions, to keep all women virally suppressed.

**Author affiliations**
[1]HIV Prevention Research Unit, South African Medical Research Council, Cape Town, South Africa
[2]Division of Epidemiology and Biostatistics, University of Stellenbosch, Cape Town, South Africa
[3]Biostatistics Unit, South African Medical Research Council, Cape Town, South Africa
[4]Centre for HIV and STI, National Institute for Communicable Diseases, Johannesburg, South Africa
[5]Faculty of Health and Life Sciences, Department of Pharmacology, University of Liverpool, Liverpool, UK
[6]Infectious Diseases Institute, Makerere University College of Health Sciences, Kampala, Uganda
[7]Blizard Institute, Queen Mary University of London, London, UK
[8]Zvitambo Institute for Maternal and Child Health Research, Harare, Zimbabwe
[9]Centre for International Health, Universitetet i Bergen, Bergen, Norway
[10]Pathogenesis and Control of Chronic and Emerging Infections, University of Montpellier INSERM, Montpellier, France
[11]CHU, Montpellier, Montpellier, France
[12]Etablissement Français du Sang, Antilles University, Paris, France

**Acknowledgements** Authors would like to thank: the Ehlanzeni district PMTCT leadership and partners for supporting the placement of the study at the clinics; the Ehlanzeni local Rob Ferreira laboratory for handling of blood samples; the staff at NICD for conducting laboratory assays for viral load measurement; the SAMRC data collection and coordinating team; and Dr Witness Chirinda for involvement in study planning discussions and contributing to data cleaning.

**Contributors** Study conceptualisation and design were performed by AEG, CW, AJP, CJL, NKN, PVdP and TT. Data analyses was done by NKN and CJL. NKN and TEM prepared the first draft of the manuscript. AP was the laboratory assays provider. All authors made critical review and revision of subsequent drafts, and gave approval of the final version. NKN and AEG are the guarantor of the work.

**Funding** This study was funded by the National Department of Health – South Africa. Authors NKN & AEG were supported by the South African Medical Research Council. Author TEM was supported by the National Research Foundation – South Africa. AJP is funded by Wellcome (grant 108065/Z/15/Z). CW is funded by Wellcome 222075/Z/20/Z. The views expressed in this manuscript are our own and not an official position of the institution or funder.

**Competing interests** None declared.

**Patient consent for publication** Not applicable.

**Ethics approval** This study involves human participants and was granted by the South African Medical Research Council ethics committee (EC002-2-2019) and the Regional Committee for Medical and Health Research Ethics West, Norway (REK-Vest no 2019/773). Participants gave informed signed consent to participate in the study before taking part.

**Provenance and peer review** Not commissioned; externally peer reviewed.

**Data availability statement** Data are available upon reasonable request. The dataset supporting the conclusions of this article will be available after the primary objectives for the study have been published. Interested researchers should contact the corresponding author to confirm if data have become open access.

**ORCID iD**
Nobubelo Kwanele Ngandu http://orcid.org/0000-0001-8883-3821

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
