## [Reviewer comments · BMJ Open]

ARTICLE DETAILS

TITLE (PROVISIONAL)	HIV viral load non-suppression and associated factors among pregnant and postpartum women in rural northeastern South Africa: a cross-sectional survey
AUTHORS	Ngandu, Nobubelo; Lombard, Carl; Mbira, Thandiwe; Puren, Adrian; Waitt, Catriona; Prendergast, Andrew; Tylleskär, Thorkild; Van de Perre, Philippe; Goga, Ameena

VERSION 1 – REVIEW

REVIEWER	Joseph, Jessica Clinton Health Access Initiative
REVIEW RETURNED	04-Nov-2021

GENERAL COMMENTS	It's great to see more work being done on VL monitoring among pregnant and postpartum women. I think these findings are important to disseminate, but would like to have a few things addressed first. 1. Your abstract does not follow the conventional headings of background/methods/results/conclusions. Please check with the journal to make sure your lack of a background in abstract and the sub-headings for methods are acceptable.2. Your main manuscript is missing an objectives statement; however, the objectives statement in the abstract is not accurate—you did not assess population risk of vertical transmission (nor can you, as your study wasn't designed for this)—you examined risk factors of mVL non-suppression3. The term “enrolled” is more often used for prospective cohort studies. Since this is cross-sectional, I would replace with “participated” or something else throughout.4. Statistical analysis: your data is clustered at the health-facility level; as such, chi-square tests and logistic regression are not appropriate as they do not account for any clustering. There are several ways you can address the clustering; this needs to be addressed and analyses re-run.5. In both your abstract and main results section, you first show which variables are associated with increased odds of mVL non-suppression, and then show which ones are associated with reduced odd (e.g. older age). To simplify, I would suggest changing your reference group so you can show all factors that increase—e.g. younger age.6. Reference #1: Is there not more recent data published? This citation only shows MTCT through 2017. And “high” is subjective, and not something mentioned in this article. (in fact, they say progress in preventing MTCT has been dramatic in the decreases that have been seen).7. Sample size calculation: it's unclear what is meant by a “district-level sample size”...as mentioned above, since your data is
---

	clustered you would have needed to add in a design effect to adjust for this. Why was a 90% confidence level used and not 95%? What were the implications to the point estimate and its precision given your actual sample size? 8. Infant HIV Status: What happened if infant status was unknown, and they had missed a testing point according to national guidelines? Did you make sure they were referred for testing or better yet test them yourselves? 9. Results: make clear that the 55 women with insufficient plasma were excluded from analyses. 10. Table 1: the numbers for Study groups seem to be reported wrong. 11. Table 2: you present row percentages but in the text write out column percentages; please align. (I would recommend switching Table 2 to show column percentages, unless you have a good reason for wanting row percentages) a. I'm very surprised that Timing of HIV-positivity is so different than ART duration, given there's only 2 women different in ≥ 1000 and 6 in < 1000. Does it have to do with the weighting and #s by group? I would have expected Timing to be significant as well. (I would also have expected the 2 variables to be collinear, so I would check that too, and if they are, remove one from your multivariable model). b. So there is no difference in Infant positivity rates between suppresses and non-suppressed women? That's a bit surprising and I'd want to dig in a bit deeper here. What study group were these women in? (ie, is it possible that the women in the earlier study group in non-suppressed haven't had time yet to have positive infants?) any other thoughts that could explain why the suppressed group have so many HIV-positive infants? 12. What were the first-line ART regimens women were on? Did you see if any 1 regimen in particular had increased odds of unsuppressed VL? 13. Another limitation that should be mentioned is the study-design. Cross-sectional studies only show one snapshot in time meaning causality can't be proven. Also the lack of randomization means that results may not be generalizable.
--	--

REVIEWER	Mandelbrot, Laurent Hopital Universitaire Louis-Mourier - APHP, Maternité
REVIEW RETURNED	02-Jan-2022

GENERAL COMMENTS	bmjopen-2021-058347 HIV viral load non-suppression and associated factors among pregnant and postpartum women in rural northeastern South Africa This is a well-done study by an eminent group of researchers in the field of PMTCT. It is of importance for public health interventions in a region where the prevalence of HIV infection is particularly high and resources are low despite a policy in South Africa of free access to antiretroviral therapy. The persistence of mother-to-child transmission during pregnancy/delivery and during breastfeeding is of major concern. 1. The authors should be commended for putting objectively into perspective the strengths and weaknesses of this study and what it does or does not add to previous knowledge. 2. The rationale for the study is the observation that the last 95% of virological suppression is not routinely monitored, despite
--

	previous surveys indicating that adherence and retention in care are major challenges. Surprisingly, the study seems to indicate that adherence challenges do not stand out, although there is an association with young age and non-disclosure which might be associated with adherence issues. 3. As acknowledged, it is a cross-sectional study of women who were selected by attending antenatal or postnatal visits, being biological mothers and consenting to respond to a detailed interview. This population of women who are engaged in care is thus likely to overestimate the proportion with virological success, as acknowledged. 4. The issue of attrition in care does not stand out in the results, but this also may be due to the study design. 5. Only 71.5% [95% CI: 63.7-78.2] had undetectable mVL, 14.7% had mVL non-suppression while 13.8% had low level viraemia (VL 50-1000 copies/mL). The definition of virological success as acknowledged was < 1000 copies since this was in a period before aligning the definition with the standard of < 50 copies/mL, but the results are also available with a 50-copy cutoff. 6. Is there information on the exact antiretroviral therapies used ? The authors might mention this as a weakness, and as well discuss the management at the time of the study. Was the first-line therapy efavirenz-based ? Thus, might the relatively poor results of the first line within the first 12 months reflect the relatively poor efficacy of this drug and the accumulation of resistance mutations in this population ? 7. Maternal plasma viral load was determined for the study. How were the results given to the patients and made available to the caregivers ? 8. The conclusion that more frequent mVL monitoring and rapid turnover is of major importance. Although the study does not directly substantiate this conclusion, it does show that progress must be made to achieve virological success in pregnant and postpartum women in this setting.
--	--

VERSION 1 – AUTHOR RESPONSE

Reviewer: 1

Ms. Jessica Joseph, Clinton Health Access Initiative

Comments to the Author:

It's great to see more work being done on VL monitoring among pregnant and postpartum women. I think these findings are important to disseminate, but would like to have a few things addressed first.

1. Your abstract does not follow the conventional headings of background/methods/results/conclusions. Please check with the journal to make sure your lack of a background in abstract and the sub-headings for methods are acceptable.

Response: The presentation of the abstract follows the BMJ-Open journal's preferred layout outlined in instructions for authors found here -> Authors - BMJ Open.

2. Your main manuscript is missing an objectives statement; however, the objectives statement in the abstract is not accurate—you did not assess population risk of vertical transmission (nor can you, as your study wasn't designed for this)—you examined risk factors of mVL non-suppression

Response: Thank you for this comment. The objective has been revised in the Abstract to: "We aimed to measure the prevalence of maternal HIV viral load (mVL) non-suppression and assess associated factors, to evaluate progress towards UNAIDS targets" (see page 2),

AND the expanded version properly labelled at the end of the main Introduction section as follows:
“Study objective: We identified one of the hotspot districts with high maternal HIV prevalence and high MTCT in South Africa and conducted a cross-sectional evaluation of the prevalence of mVL non-suppression (VL>1000 copies/mL) and associated factors during peripartum and postpartum periods, to evaluate progress towards the UNAIDS targets.” (see page 5)

3. The term “enrolled” is more often used for prospective cohort studies. Since this is cross-sectional, I would replace with “participated” or something else throughout.

Response: The word “enrolled” previously used in the abstract under ‘Participants’ section has been replaced with “recruited” or “participated”. (see page 2)

4. Statistical analysis: your data is clustered at the health-facility level; as such, chi-square tests and logistic regression are not appropriate as they do not account for any clustering. There are several ways you can address the clustering; this needs to be addressed and analyses re-run.

Response: We notice the study design was not clear in the methods and hence the health facilities were mistaken for clusters. In the study design, the health facilities were not clusters but were the primary sampling units used to collect data for all the five strata (strata being the different phases of the PMTCT cascade). A stratified survey-based design was used to combine the five strata into a single analysis. Survey sampling weights were applied and calculated from the inverse of the proportion of the sample size realized at each primary sampling unit (health facility). The section on statistical methods has been updated accordingly to make this clear (page 7) and a sentence added to the Study design section on page 5.

5. In both your abstract and main results section, you first show which variables are associated with increased odds of mVL non-suppression, and then show which ones are associated with reduced odd (e.g. older age). To simplify, I would suggest changing your reference group so you can show all factors that increase—e.g., younger age.

Response: We have re-arranged age-group and used the oldest group as the reference in the regression models (see Abstract results, main text results on page 15 and Table 3 on page 15-16). However, we have not re-arranged BMI given that the highest group has uncommonly high values and the observed results cannot be explained by the study data. Hence using the ‘normal BMI’ range as the reference group makes most methodological sense in this case.

6. Reference #1: Is there not more recent data published? This citation only shows MTCT through 2017. And “high” is subjective, and not something mentioned in this article. (in fact, they say progress in preventing MTCT has been dramatic in the decreases that have been seen).

Response: We agree that incidence of MTCT has declined but it has not reached HIV elimination targets. The subjective word “high” has been replaced with “public health concern”. Reference #1 has been updated and replaced with the most recent data published online (<https://data.unicef.org/topic/hivaids/emtct/#more--1656>) in ‘July 2021’. (top of page 4).

7. Sample size calculation: it’s unclear what is meant by a “district-level sample size”...as mentioned above, since your data is clustered you would have needed to add in a design effect to adjust for this.

Response: Please see response to comment 4, above.

7b: Why was a 90% confidence level used and not 95%?

Response: We deemed the precision of 0.05 to be more important. As stated in the Methods: “A district-level sample size was calculated using an assumed mVL non-suppression prevalence of 25% and a precision of 5% at a 90% confidence level”

7c: What were the implications to the point estimate and its precision given your actual sample size?

Response: Given the observed proportion of participants with VL>1000 copies/mL to be= 0.147, the minimum sample size achieving a precision of 0.05 is N=136. The actual sample size at sub-group level ranged between 75 and 176. Therefore, the precision was slightly reduced in some sub-groups – ranging between 0.05 and 0.068 (i.e., for sample of 176 (precision = 0.05), 128 (0.0514), 123 (0.0526), 110 (0.056) and 75 (0.068). This information has been added to the beginning of the main Results section after the sentence of realized sample sizes as follows: “The precision for these achieved samples remained close to 0.05 and was 0.05, 0.0514, 0.068, 0.0526 and 0.056, respectively”. (page 8 first paragraph of the Results section).

8. Infant HIV Status: What happened if infant status was unknown, and they had missed a testing point according to national guidelines? Did you make sure they were referred for testing or better yet test them yourselves?

Response: If an infant had an unknown HIV status, blood samples were only taken from those infants who had no blood draws at the clinic within the past 4 weeks. All study results were given to the facility managers to provide them to patients at follow-up visits.

9. Results: make clear that the 55 women with insufficient plasma were excluded from analyses.

Response: The following sentence has been added at the end of the first paragraph of the main Results section: “The 55 women with insufficient plasma were excluded from the analysis.” (page 8)

10. Table 1: the numbers for Study groups seem to be reported wrong.

Response: Thank you for identifying this error, the correct sample size numbers have been added to Table 1. (page 9)

11. Table 2: you present row percentages but in the text write out column percentages; please align. (I would recommend switching Table 2 to show column percentages, unless you have a good reason for wanting row percentages)

Response: Table 1 column percentages were used to describe the overall study sample regardless of mVL status. Table 2 describes mVL prevalence by risk factors – hence the row percentages. The percentages are used correctly in the text depending on which Table is referenced for that text, and what the aim of the analysis is.

a. I'm very surprised that Timing of HIV-positivity is so different than ART duration, given there's only 2 women different in ≥ 1000 and 6 in < 1000 . Does it have to do with the weighting and #s by group? I would have expected Timing to be significant as well. (I would also have expected the 2 variables to be collinear, so I would check that too, and if they are, remove one from your multivariable model).

Response: We have investigated for collinearity between ART duration and timing of HIV positivity and found a variance inflation factor of 6.3 and decided to exclude timing of HIV positivity, based on VIF cutoff of 5. The VIF may not be extreme because for study strata at postpartum stages >6months to 24 months – ART duration of '>12 months' does not necessarily mean 'before pregnancy'. Nonetheless the overall picture of the regression results remains the same and the previous significant and non-significant associations have not changed. Minor changes in point estimates and p-values have been edited accordingly. See results section “Factors associated with maternal viral load non-suppression” pages 14 – 15 and Table 3 and the abstract-results section.

b. So there is no difference in Infant positivity rates between suppressed and non-suppressed women? That's a bit surprising and I'd want to dig in a bit deeper here. What study group were these women in? (ie, is it possible that the women in the earlier study group in non-suppressed haven't had time yet to have positive infants?) any other thoughts that could explain why the suppressed group have so many HIV-positive infants?

Response: The sample size was powered to report the prevalence mVL non-suppression and hence was not sufficient to accurately estimate MTCT which generally has a much lower estimate in comparison. Most importantly, the cross-sectional nature of the study design cannot always show the difference in MTCT between suppressed and non-suppressed women, given we do not know the duration of non-suppression versus breastfeeding without sufficient antiretroviral coverage. We have added this as an example of the limitations of this cross-sectional study as follows: "An example of the limitation of a cross-sectional design in this context is not being able to understand timing of infant HIV testing and the relationship between infant HIV-positivity versus mVL given there is no information on duration of non-suppression in the mother against attrition in care, interruptions in ART intake and feeding practices." (page 21)

12. What were the first-line ART regimens women were on? Did you see if any 1 regimen in particular had increased odds of unsuppressed VL?

Response: The options for ART regimen in the public sector are limited and not much variation is expected for persons receiving the same regimen. For example, the first line regimen was a fixed-dose combination of TDF + 3TC (or FTC) + EFV. Women with contraindications were recommended AZT or where feasible referred to a doctor to assess and prescribe one of these drugs. Nonetheless, in this study we attempted to gather the specific drugs but the data not available. Even though most women knew they were taking a fixed-dose combination (N=525), information on specific drugs was only available from 93 participants as follows: 3TC N=6, LPVr N=12, TDF N=12, FTC N=8, EFV N=55, DTG N=0. Therefore, the data are not sufficient to disentangle the effects of specific ART drugs. Given your comment, we have added the following to the limitations: "The possible influence of specific ART drugs could not be assessed due to lack of data on specific drug combinations used. This, however, may have minor effects given that options for the first line regimen were limited to efavirenz-based regimens, hence not much diversity in specific drugs used [22]". (page 20-21). The regimens used during the time of this study have also been explained in the methods section under 'ART-related factors' (page 6).

13. Another limitation that should be mentioned is the study-design. Cross-sectional studies only show one snapshot in time meaning causality can't be proven. Also the lack of randomization means that results may not be generalizable.

Response: We have clearly mentioned that the study design has limitations. An example arising from comment 11b above has also been added. (page 21). Randomization does not apply to this study design. In the study objective (end of the Introduction section), we indicate that the district was purposively selected based on MTCT performance indicators. We do not assume or imply that this study is generalizable to other settings. Such purposive selection precludes generalizability.

Reviewer: 2

Dr. Laurent Mandelbrot, Hopital Universitaire Louis-Mourier - APHP

Comments to the Author:

bmjopen-2021-058347

HIV viral load non-suppression and associated factors among pregnant and postpartum women in rural northeastern South Africa

This is a well-done study by an eminent group of researchers in the field of PMTCT. It is of importance for public health interventions in a region where the prevalence of HIV infection is particularly high and resources are low despite a policy in South Africa of free access to antiretroviral therapy. The persistence of mother-to-child transmission during pregnancy/delivery and during breastfeeding is of major concern.

1. The authors should be commended for putting objectively into perspective the strengths and weaknesses of this study and what it does or does not add to previous knowledge.

Response: Thank you.

2. The rationale for the study is the observation that the last 95% of virological suppression is not routinely monitored, despite previous surveys indicating that adherence and retention in care are major challenges. Surprisingly, the study seems to indicate that adherence challenges do not stand out, although there is an association with young age and non-disclosure which might be associated with adherence issues.

Response: We used adherence during the past 7 days as a proxy for overall adherence to minimize recall bias that often arises when asking for longer recall periods. However, 7-day recall has its limitations in that it might not reflect what happened during the entire period since the last contact with the clinic. In the paper we had mentioned desirability bias in reporting adherence and we have also now added this limitation as follows: "Use of a 7-day recall for adherence to ART, although limiting recall bias, might suffer from low representativeness of the entire period since the last clinic contact." (page 20).

3. As acknowledged, it is a cross-sectional study of women who were selected by attending antenatal or postnatal visits, being biological mothers and consenting to respond to a detailed interview. This population of women who are engaged in care is thus likely to overestimate the proportion with virological success, as acknowledged.

Response: Agreed, thank you for emphasizing this point.

4. The issue of attrition in care does not stand out in the results, but this also may be due to the study design.

Response: We have added attrition to the example of the limitation of a cross-sectional study design, as follows: "An example of the limitation of a cross-sectional design in this context is not being able to understand the relationship between infant HIV-positivity versus mVL given there is no information on duration of non-suppression in the mother against attrition in care, interruptions in ART intake and feeding practices." (page 21)

5. Only 71.5% [95% CI: 63.7-78.2] had undetectable mVL, 14.7% had mVL non-suppression while 13.8% had low level viraemia (VL 50-1000 copies/mL). The definition of virological success as acknowledged was < 1000 copies since this was in a period before aligning the definition with the standard of < 50 copies/mL, but the results are also available with a 50-copy cutoff.

Response: Yes, this is correct, the additional result with a 50 cutoff is added descriptively only to inform the new guidelines as far as these data could. We elaborate on this at the end of the limitations section and have also added the following to the elaboration: "The result of VL=50-1000 copies/mL is only reported descriptively." (page 21)

6. Is there information on the exact antiretroviral therapies used? The authors might mention this as a weakness, and as well discuss the management at the time of the study. Was the first-line therapy efavirenz-based? Thus, might the relatively poor results of the first line within the first 12 months reflect the relatively poor efficacy of this drug and the accumulation of resistance mutations in this population?

Response: Yes, the first-line regimen was efavirenz-based. Given your comment, we have added the following to the Discussion: “Alternatively, it could be due to the dominance of efavirenz-based first-line regimen which has been shown to be associated with delayed viral suppression in this population [26 27] or a high population-level risk of drug resistance previously observed in South African rural populations [28], or a combination of these factors. The new guidelines currently being rolled-out offer pregnant women a dolutegravir-based regimen as a first-line choice and alternative to efavirenz regimen as part of the new efforts to improve viral load suppression[29].” (page 17)

And to the limitations: “The possible influence of specific ART drugs could not be assessed due to lack of data on specific drug combinations used. This however, may have minor effects given the options for the first line regimen were limited to efavirenz-based regimens, hence not much diversity in specific drugs issued [22]”. (page 20-21)

In addition we have described the ART regimens used during this study in the methods section under “ART-related factors” (page 6).

7. Maternal plasma viral load was determined for the study. How were the results given to the patients and made available to the caregivers?

Response: Results were sent to clinic routine staff to inform participants when they returned for their routine schedules.

8. The conclusion that more frequent mVL monitoring and rapid turnover is of major importance. Although the study does not directly substantiate this conclusion, it does show that progress must be made to achieve virological success in pregnant and postpartum women in this setting.

Response: Thank you. Regimen switching and alternative ART regimens added to the new guidelines have also been mentioned in the conclusion statements in the abstract (page 2) and main text (page 21).

Other revisions made

We have replaced the use of “HIV-infected women” with “women living with HIV, (WLHIV)” throughout the manuscript.

The Policy guidelines around viral load testing, HIV testing and ART regimens have been described in the methods section to put the results into context. (pages 6)

Funding received by author CW has been included. (page 22)

VERSION 2 – REVIEW

REVIEWER	Joseph, Jessica Clinton Health Access Initiative
REVIEW RETURNED	11-Feb-2022
GENERAL COMMENTS	I'm very pleased with the reviewer's responses and believe they have done their due diligence and beyond. I think all the edits they incorporated into the text have strengthened this manuscript greatly. The only outstanding comment I have is that for my review comment #3, regarding the change of 'enrolment' to 'recruitment' or 'participant', I'd love to see it changed throughout as well. There are 4-5 other instances where "enrolment" could be changed to "recruitment", including in the tables.